# On the Worst Prompt Performance of Large Language Models*

**Bowen Cao**$^{\diamond,\dagger}$  **Deng Cai**$^{\heartsuit,\ddagger}$  **Zhisong Zhang**$^{\heartsuit}$  **Yuexian Zou**$^{\spadesuit}$  **Wai Lam**$^{\diamond}$

$\diamond$ The Chinese University of Hong Kong

$\heartsuit$ Tencent AI Lab  $\spadesuit$ Peking University

bwcao@link.cuhk.edu.hk, thisisjcykcd@gmail.com, zhisonzhang@tencent.com

zouyx@pku.edu.cn, wlam@se.cuhk.edu.hk

## Abstract

The performance of large language models (LLMs) is acutely sensitive to the phrasing of prompts, which raises significant concerns about their reliability in real-world scenarios. Existing studies often divide prompts into task-level instructions and case-level inputs and primarily focus on evaluating and improving robustness against variations in tasks-level instructions. However, this setup fails to fully address the diversity of real-world user queries and assumes the existence of task-specific datasets. To address these limitations, we introduce ROBUSTAL-PACAEVAL, a new benchmark that consists of semantically equivalent case-level queries and emphasizes the importance of using the worst prompt performance to gauge the lower bound of model performance. Extensive experiments on RO-BUSTALPACAEVAL with ChatGPT and six open-source LLMs from the Llama, Mistral, and Gemma families uncover substantial variability in model performance; for instance, a difference of 45.48% between the worst and best performance for the Llama-2-70B-chat model, with its worst performance dipping as low as 9.38%. We further illustrate the difficulty in identifying the worst prompt from both model-agnostic and model-dependent perspectives, emphasizing the absence of a shortcut to characterize the worst prompt. We also attempt to enhance the worst prompt performance using existing prompt engineering and prompt consistency methods, but find that their impact is limited. These findings underscore the need to create more resilient LLMs that can maintain high performance across diverse prompts. Data and code are available at https://github.com/bwcao/RobustAlpacaEval.

## 1 Introduction

In recent years, large language models (LLMs) have made extraordinary progress (Brown et al., 2020; Touvron et al., 2023; Jiang et al., 2023; Team et al., 2024) and one key factor in their success is their ability to adapt to diverse tasks through prompting. However, the performance of these LLMs is significantly sensitive to the prompts they receive (Gu et al., 2022; Mizrahi et al., 2024; Sun et al., 2023; Li et al., 2024). Even minor alterations in format, without semantic changes, can trigger substantial performance degradation (Sclar et al., 2023). Consequently, prompt engineering has surfaced as a critical component, playing a critical role in unlocking the full potential of these models (Shin et al., 2020; Zhou et al., 2022a; Pryzant et al., 2023; Prasad et al., 2023; Gonen et al., 2022; Schulhoff et al., 2024).

---

*The work described in this paper is partially supported by a grant from the Research Grant Council of the Hong Kong Special Administrative Region, China (Project Code: 14200719).

†Work done during an internship at Tencent AI Lab.

‡Corresponding author.

|  | **Existing Work** | **Real-world User Queries** |
|---|---|---|
| **Task-level Instruction** | Assume you are a customer service representative. Please provide customer service to a customer regarding their difficulties in accessing a website. | Case1: As a customer service representative, kindly assist a customer who is facing challenges with our website, specifically reporting an inability to sign in over the last four days. |
| **Case-level Inputs** | **Case1:** The customer states that they have been unable to sign in for the past four days.
**Case2:** A customer reports that they are unable to complete an online purchase due to the website's checkout process consistently failing.
…… | Case2: Craft a narrative that brings to life a character sketched as a 5-year-old boy with an insatiable curiosity and a penchant for questioning, while also harboring a strong aversion to adhering to rules. Feel free to elaborate on his traits and weave in incidents that highlight his personality. |

Figure 1: An example illustrating the gap between existing benchmarks that evaluate prompt consistency and real user queries.

Despite the efficacy of prompt engineering, it is not without its drawbacks. First, it is unrealistic to expect users to master the art of designing optimal prompts or to invest a significant amount of time in doing so. Second, automatic prompt engineering often necessitates testing on substantial labeled data to pinpoint the most effective prompts (Shi et al., 2022; Prasad et al., 2023), a process that is impractical for users who just have one or a few unlabeled queries. Therefore, investigating LLMs' robustness to prompt variances is of great research and practical value.

Previous research on prompt robustness (Mizrahi et al., 2024; Sclar et al., 2023; Sun et al., 2023) divides a user query into two parts: *task-level instruction* and *case-level input*. The task-level instruction is an abstract task definition and explanation supplemented by case-level concrete input (See the left part of Figure 1). This setup is reminiscent of conventional NLP practice, where task-specific models are developed using a testing set for each well-defined NLP task. Consequently, they have focused on the LLM's resilience to task-level instructions exclusively, reporting the average performance across testing set cases. Nonetheless, these studies have limitations. First, they disregard that the best task-level instruction might vary across individual cases. Second, they overlook the impact of variations in case-level input on model performance. Last but not least, real-world user queries often do not explicitly segregate task-level instruction and case-level input (See the right part of Figure 1). These queries may cover a wide array of tasks and it is not possible to optimize the prompts through evaluating on a task-specific testing set.

In this paper, we present a comprehensive study that goes beyond the conventional approach of evaluating LLMs. We shift the focus from task-level instructions to diverse real-world user queries. Our work introduces a new benchmark, ROBUSTALPACAEVAL, that includes semantically equivalent case-level queries across various tasks, offering a more holistic analysis. We argue that the *worst prompt performance*, defined as the lowest performance a model exhibits across different paraphrases of a query with equal semantics and fluency, is a crucial metric for assessing the lower bound of LLM performance. Our extensive experiments on ChatGPT (gpt-3.5-turbo-1106) and six open-source LLMs from the Llama, Mistral, and Gemma families reveal substantial variability in model performance. For instance, Llama-2-70B-chat model shows a difference of up to 45.48 points in win-rate against GPT4 using RobustAlpacaEval, and its worst prompt performance can be as low as 9.38%. Our findings further reveal that the worst prompts, which cause the model to perform the worst in specific cases, are not universally applicable across different models. Each model also exhibits unique preferences towards all paraphrases, as demonstrated by their inconsistent performance rankings on identical cases. Beyond the absence of model-agnostic traits in identifying the worst prompts, model-dependent prompt features, such as perplexity and hidden states, also prove inadequate in forecasting model performance. Furthermore, we also find that different models are more sensitive in different cases, which further underscores the complexity involved in tackling the issue of worst prompt performance.

In summary, the contributions of this paper can be summarized as follows:

• We introduce a new benchmark, ROBUSTALPACAEVAL, that encompasses semantically equivalent queries of diverse real-world tasks, offering a more holistic assessment compared to existing benchmarks that focus solely on rephrasing task-level instructions.

• Through extensive experiments on ChatGPT and six open-source LLMs from the Llama, Mistral, and Gemma families, we unveil significant variability in model performance and highlight the difficulty in predicting prompts that lead to the worst performance.

• We show the shortcomings of most existing methods in improving the worst prompt performance of models. Yet a voting-based method effectively enhances the stability of model performance.

## 2   Related Work

Existing research on prompt robustness can be classified into two categories. On the one hand, research efforts have been focused on enhancing the inherent resilience of LLMs to prompt variations, namely *prompt consistency*. On the other hand, progress has been made in automating *prompt engineering*, the process to find the optimal prompt that yields the best performance.

**Prompt Consistency.**   Previous studies (Gu et al., 2022; Wang et al., 2023a; Zhu et al., 2023; Wang et al., 2023b; Salinas and Morstatter, 2024) have explored the robustness of LLMs to intentional perturbations such as word deletion and sentence shuffling. Our work deviates from these perturbation-based studies, as we are interested in model performance across semantically equivalent and syntactically fluent prompts. While it is anticipated that a flawed prompt would lead to performance decline, we do not expect such performance variations in our setting.

Recent research (Mizrahi et al., 2024; Sclar et al., 2023; Sun et al., 2023) has also examined the variability in model performance with semantically equivalent prompts. However, these studies solely focused on task-level instructions and overlooked the variations within case-level inputs. Moreover, the models used in these studies are trained and evaluated on traditional NLP datasets, while today's LLMs are predominantly instruction-tuned on and serve diverse user prompts. This discrepancy suggests that the findings of these studies may not fully apply to the latest generation of LLMs.

**Prompt Engineering.**   A number of studies utilize gradient-based methods for prompt optimization (Shin et al., 2020; Shi et al., 2022; Li and Liang, 2021; Qin and Eisner, 2021; Lester et al., 2021; Liu et al., 2023). However, their dependence on substantial labeled training data and gradient computing limits their applicability for users with only a few unlabeled queries or gradient-less APIs. Furthermore, these methods incur significant computational costs as the model scale increases.

Conversely, gradient-free methods aim to find the optimal prompt through exploration and scoring (Prasad et al., 2023; Zhou et al., 2022a; Pryzant et al., 2023; Chen et al., 2023; Yang et al., 2023). During the exploration phase, they create analogous prompts using techniques such as rephrasing the current prompts with an LLM. Subsequently, the best prompts are selected based on the model's performance on downstream tasks. This process can be iterative. Although these approaches can proficiently pinpoint prompts that result in better performance for a specific task, they all necessitate a test set for prompt scoring. However, in real-world scenarios, such test sets are often absent. Gonen et al. (2022) suggests that prompt perplexity can be an effective data-free criterion. However, our experiment results tell a different story, indicating that prompt perplexity is not a reliable metric.

## 3   Benchmarking the Worst Prompt Performance

We present the construction process of our new benchmark, ROBUSTALPACAEVAL (§3.1) and report the results on ChatGPT and six open-source LLMs from Llama, Mistral, and Gemma families.

### 3.1   A New Benchmark: ROBUSTALPACAEVAL

**Data.**   Our benchmark is based on TinyAlpaceEval (Polo et al., 2024), which is a condensed subset of the AlpacaEval (Li et al., 2023) benchmark, created to enable efficient assessment of LLMs. We develop ROBUSTALPACAEVAL by creating ten[4] paraphrases for each query within TinyAlpacaEval. To save manual efforts, this is first accomplished automatically through GPT4. Subsequently, each paraphrase is manually reviewed and revised to ensure semantic integrity and human-like fluency. We

---

[4]We find that 10 paraphrases are sufficient to steadily reflect the variability in the model's performance. Further details can be found in Appendix A.1.

| Model | Orig. Perf. ↑ | Worst Perf. ↑ | Best Perf. ↑ | Avg. Perf. ↑ | Standard Dev. ↓ |
|---|---|---|---|---|---|
| Gemma-1.1-2b-it | 16.32 | 4.42 | 36.60 | 15.27 | **11.78** |
| ChatGPT | 17.46 | 5.44 | 39.88 | 19.96 | 12.86 |
| Mistral-7b-instruct | 24.56 | 4.22 | 45.26 | 21.82 | 14.60 |
| Llama-2-7b-chat | 25.61 | 5.42 | 43.54 | 19.52 | 13.32 |
| Llama-2-13b-chat | 27.48 | 4.83 | 52.05 | 23.97 | 16.25 |
| Gemma-1.1-7b-it | 29.57 | 8.73 | **62.38** | **31.04** | 19.07 |
| Llama-2-70b-chat | **32.23** | **9.38** | 54.86 | 29.18 | 15.61 |

Table 1: Results on our RobustAlpacaEval benchmark. The model order is arranged according to their original performance. The substantial range between the worst and best performance suggests the robustness issues in LLMs' instruction-following ability. Scaling up model sizes, while improving average performance, does not enhance robustness.

use carefully crafted prompts to promote the diversity of paraphrases, which is evidenced by the fact that the average of length-normalized edit distance[5] between each pair of paraphrases is 0.7234 at the word level. The few-shot paraphrasing instruction utilized in this process is detailed in Appendix A.2.

**Metrics.** In line with common practice, we use weighted win-rate (Li et al., 2023) as our performance metric; It uses an evaluator to compare the output of the target model against that of a reference model, and estimates the winning probability of the target model. Specifically, we employ the gpt4_turbo model as the evaluator and the reference model. We term the model's performance on the original prompt as *original* performance. We also report the *worst*, *best*, *average* performances across all eleven prompts as well as the standard deviation. For each metric, we average the results across all cases in RobustAlpacaEval.

### 3.2 Results

The results shown in Table 1 reveal several key findings:

**There is a significant gap between the worst performance (lower bound) and best performance (upper bound) for all models.** For instance, the worst and best performance of Llama-2-70B-chat are 0.094 and 0.549, respectively, indicating a difference of 0.455. This suggests that the current LLMs' ability to follow instructions is not robust enough. Even instructions with identical semantics and fluent expressions could lead models like Llama2-70B-chat to plummet from a level comparable to GPT4 (0.5 indicates equivalence to the reference model) to far below the average level (0.292).

**While scaling up models enhances their ability to follow instructions, it does not correspondingly increase their robustness.** For example, the average performance of Llama-2-7B/13B/70B-chat shows a marked improvement, rising from 0.195 to 0.24, and finally to 0.292, but their robustness (indicated by the Standard Dev.) exhibits a slight decline, recorded at 0.133, 0.163, and 0.156 respectively. A similar trend can be observed with the Gemma family models. Despite Gemma-7b outperforming the 2b model with an average performance of 0.31 compared to 0.153, its robustness is inferior, registering at 0.191 compared to the 2b model's 0.118.

**The original performance assessment only provides a narrow perspective of a model's overall performance.** Specifically, the original performance serves as a reliable metric for ranking within the same model family, as it aligns well with the best, worst, and average performances. However, we find that this alignment weakly correlates when comparing across different model families, such as Gemma and Llama, questioning the methodological validity of using just one type of phrasing for a query to assess performance. Furthermore, the original performance does not fully encapsulate a model's potential, including the boundary and average of its performance, and fails to reflect the stability of the model's performance.

---

[5]We compute the edit distance between prompts $x$ and $y$ normalized by their average length as: $2 * \text{EditDistance}(x, y) / (|x| + |y|)$

# 4 Identifying the Worst Prompts

Given the noticeable performance disparities across semantically equivalent prompts, our next question is: can we identify the worst prompt among these paraphrases? This would not only aid our understanding of the model's lower bound but also be instrumental in improving model performance by guiding the refinement of prompts. We investigate this matter from both model-agnostic (§4.1) and model-dependent (§4.2) perspectives.

## 4.1 Model-agnostic Analysis

We examine the model-agnostic attributes of the worst prompts from two perspectives: (*i*) we assess whether the worst prompts overlap across diverse models, and (*ii*) we probe whether the rankings of prompts are consistent across different models.

**Overlap of the worst prompts across different models.** If many of the worst prompts are model-agnostic, there must exist certain prompts that, among their semantically equivalent paraphrases, consistently rank as the worst performing across all models. To quantify the prevalence of such prompts, we calculate the rate of the model-agnostic worst-$k$ prompt. Let $W_m(x, k)$ denote the worst-$k$-performing prompts for model $m$ on case $x$, and let the set of worst-$k$ prompts for model $m$ on dataset $D$ be $W_m(D, k) = \bigcup_{x \in D} W_m(x, k)$. We calculate the overlap rate of the worst-$k$ prompts for the tested models, which ranges from 0 to 1, as follows:

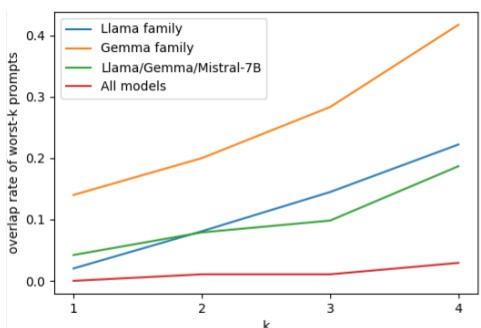

Figure 2: The overlap rate of model-agnostic worst-$k$ prompts across different models. The low result indicates a minimal occurrence of universally poor prompts.

$$R(M, k, D) = \frac{\bigcap_{m \in M} W_m(D, k)}{k * |D|} \quad (1)$$

As shown in Figure 2, the overlap rate of the worst-$k$ prompts for Llama/Gemma family models is noticeably higher than that for all models, indicating a stronger consistency among models from the same family. However, even so, when $k = 1$ (considering only the worst prompt), the metric for Llama family models is only 2% (13% for Gemma), and less than 10% (20% for Gemma) when $k = 2$. Not to mention the results of all models are much lower, nearly zero. These results suggest that the prompts showcasing the lower-bound performance of the models are often model-specific.

**Performance Rankings of prompts across different models.** We continue to check if the rankings among different models are consistent. We utilize Kendall's W (Kendall and Smith, 1939), commonly referred to as the coefficient of concordance, as a means to quantify the correlation among performance rankings derived from various models. Mathematically, Kendall's W is defined as follows:

$$W = \frac{12 \sum_{i=1}^{n} (R_i - \bar{R})^2}{k^2 (n^3 - n)} \quad (2)$$

where $R_i$ represents the sum of the ranks assigned to the $i$-th prompt by all models. $\bar{R}$ is the mean of total ranks, calculated as $\bar{R} = \frac{1}{n} \sum_{i=1}^{n} R_i$. $k$ is the number of models, and $n$ is the number of prompts being ranked. The value of W ranges from 0 to 1, where 0 indicates no agreement and 1 indicates complete agreement among the judges[6].

We compute the average Kendall's W across all cases as the overall consistency. The results presented in Table 2 reveal the following observations: (*i*) The rankings within the Llama and Gemma family models exhibit only moderate consistency. The slightly higher consistency of the Gemma family models could be attributed to the fewer scales (2b/7b). This suggests that individual models within the same family still possess their unique strengths and weaknesses. (*ii*) The consistency between all models is significantly lower, underscoring the difficulty in establishing a model-agnostic standard for determining "good" and "bad" prompts.

---

[6]W values less than 0.1 are often considered as no or negligible agreement; 0.1 to 0.3 suggest weak agreement; 0.3 to 0.5 indicate moderate agreement; and W values above 0.5 show strong agreement (Sidney, 1957).

| Model | Kendall's W | Agreement Levels Distribution | | | |
|---|---|---|---|---|---|
| | | Negligible | Weak | Moderate | Strong |
| Llama family | 0.443 | 0 | 0.242 | 0.414 | 0.343 |
| Gemma family | 0.548 | 0 | 0.08 | 0.28 | 0.64 |
| Llama/Gemma/Mistral-7B | 0.401 | 0.011 | 0.326 | 0.411 | 0.253 |
| All models | 0.238 | 0.053 | 0.723 | 0.202 | 0.021 |

Table 2: We report the average value of Kendall's W across all cases. We also calculate the proportion of cases with different levels of consistency. The ranges for each level of consistency are as follows: Negligible (W < 0.1), Weak ($0.1 \leq W < 0.3$), Moderate ($0.3 \leq W < 0.5$), and Strong (W $\geq$ 0.5).

**Overlap of Sensitive Cases.** The above experiment results underscore that the worst prompt is almost unpredictable in advance without given the model. Our next question is whether different models suffer from the same prompt variances. Concretely, we classify a case as a sensitive case if and only if the model's performance range (the difference between the best and worst performances) exceeds a threshold (*e.g.*, 0.5).[7] Figure 10 in Appendix B presents an example of a sensitive case. We measure the overlap of sensitive cases between different models by calculating the average Intersection over Union (IoU) of each pair of models within the tested model set. Let $S_m(D)$ represent the set of sensitive cases for model $m$ on dataset $D$. The overlap measure of the tested model set $M$ is:

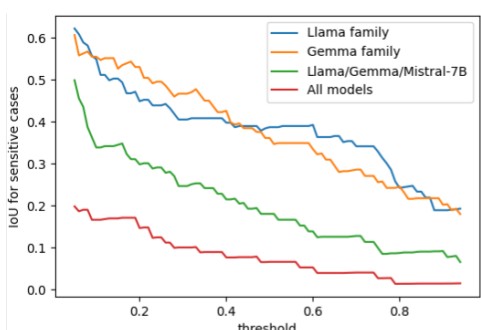

Figure 3: IoU fluctuation across varying sensitive case thresholds for diverse model sets. The IoU drops below 0.2 across all models, indicating a scarcity of model-agnostic traits.

$$IoU(M, D) = \mathbb{E}_{m_i, m_j \in M, i \neq j} IoU(m_i, m_j, D) \tag{3}$$

$$IoU(m_i, m_j, D) = \frac{S_{m_i}(D) \cap S_{m_j}(D)}{S_{m_i}(D) \cup S_{m_j}(D)} \tag{4}$$

Figure 3 illustrates the fluctuation of IoU in response to varying sensitive case thresholds across diverse model sets. The observations are as follows: (i) The IoU is markedly higher among the Llama/Gemma family models, indicating a greater commonality of vulnerabilities within the same model family. For models from different families, such as the 7B models of Llama, Gemma, and Mistral, their IoU is lower despite similar overall performance (as shown in Table 1). (ii) When considering all models, the IoU drops significantly to less than 0.2, suggesting *the absence of a clear model-agnostic characteristic*. (iii) As the threshold increases, there is a consistent decrease in the IoU across all model sets. This suggests that the most sensitive cases for a model are often unique to it, thereby emphasizing the existence of distinct issues inherent to each model.

**Discussion.** The above experiments demonstrate that the performance rankings of different prompts are inconsistent among different models. Furthermore, different models may suffer from different prompt variances. **Therefore, it is unlikely to characterize the worst prompts using model-independent features.** To this end, we omit further analyses on the relationship between model performance and linguistic features of the prompts, such as sentence length, syntactical complexity, wording choice, and paraphrasing methods.

### 4.2 Model-dependent Analysis

The findings from §4.1 suggest that we can hardly predict the prompt performance in advance without access to the model. We then turn to explore the possibilities of building the worst prompt predictors using model-aware features.

---

[7]We also attempt to define the sensitivity through standard deviation and the coefficient of variation, finding results very similar to those obtained when using range as the metric.

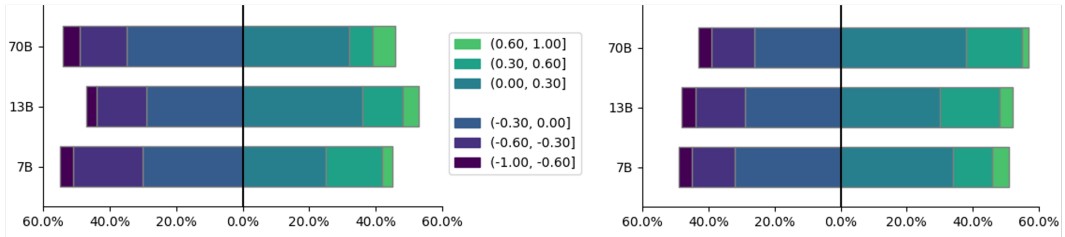

Figure 4: Distribution of Pearson correlation coefficients between model performance and prompt perplexity (left) and prompt's Min-K% Prob (right) for Llama-family models across all cases. The absolute values of correlation in the ranges of (0, 0.3], (0.3, 0.6], and (0.6, 1] respectively denote weak/no correlation, moderate correlation, and strong correlation.

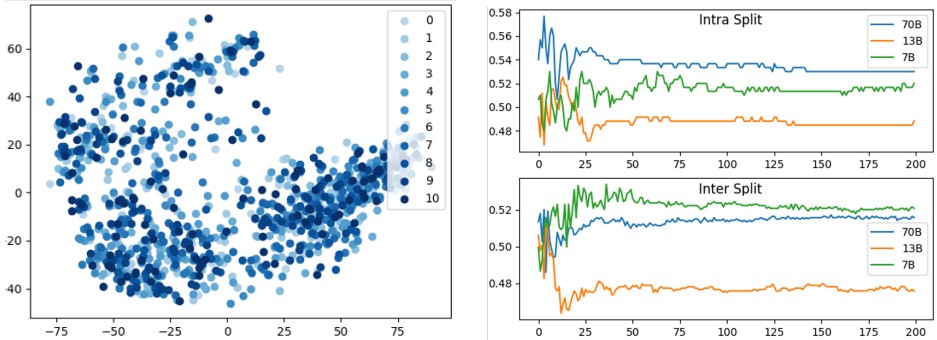

Figure 5: (a) Visualization of Llama-2-7B-chat model's hidden states using 2-dimensional PCA. The color gradient, from light to dark, represents the ranking of model performance on each case's 11 prompts, from low to high. (b) Probing Llama-2-7B-chat model's hidden states for prompt scoring. The x-axis stands for training steps. The y-axis represents the accuracy of the model's predictions, quantified as the proportion of correctly judged prompt pairs out of all test pairs.

- **Prompt Perplexity.** Gonen et al. (2022) report that lower prompt perplexity generally correlates with superior model performance. To verify this assertion, we compute the Pearson correlation coefficients between perplexity and performance across all cases. The results are illustrated in Figure 4 (left half). The layout divides correlations into negative and positive (left/right of the 0 tick), with correlation strengths categorized as weak/no (0 to 0.3), moderate (0.3 to 0.6), and strong (0.6 to 1). Analyzing the Llama family models, we note the following: (*i*) About 45% of cases show weak or no correlation. (*ii*) Positive and negative correlations are nearly equally distributed. (*iii*) The distribution of cases across different correlation degrees is fairly balanced. These results are consistent with observations from other models analyzed (detailed in Appendix B), indicating no definitive correlation between prompt perplexity and model performance.

- **Min-K% Prob.** We also explore the potential relationship between model performance and the prompt's Min-K% Prob (Shi et al., 2023), a method related to pre-training data detection, which judges the model's familiarity with the text based on the average log-likelihood of the lowest k% tokens in the sentence. Intrigued by this, we carry out an analysis akin to that for perplexity, only to find weak positive correlations that lack statistical significance, as shown in the right half of Figure 4.

- **Hidden States.** We shift our focus to investigate model perceptions of prompt quality through the lens of model representation, using: (i) Principal Component Analysis (PCA) for direct visualization of hidden states, as shown in Figure 5 (a). The color gradient reflects the model's performance ranking across 11 prompts per case, revealing no distinct separation between different performance levels. (ii) Probing technique to quantify the extent to which the model's internal representations are predictive of its performance. We train a reward model (a 3-layer MLP in practice) that scores prompts based on hidden states and apply the standard training loss function (Ouyang et al., 2022), which is defined as:

$$\mathbb{E}_{(p_g, p_b) \sim P} \left[ \log \left( \sigma \left( r_\theta(p_g) - r_\theta(p_b) \right) \right) \right] \tag{5}$$

where $P$ represents the entire set of paraphrase pairs, each pair being formed by combining two paraphrases from the same case, $r_\theta(p)$ is the output of the reward model, and $p_g$ is the preferred prompt when compared to $p_b$ in a pair. We implement two training-test set partitioning strategies: (*i*) Intra: Dividing prompts within each case into training and testing sets at a 3:1 ratio. (*ii*) Inter: Dividing all cases into training and testing sets at a 3:1 ratio. For each case in the test set, we pair up the paraphrases and determine the correctness of the model's scoring on each pair based on their performance ranking. We report the average correctness rate across all pairs as the accuracy. As shown in Figure 5 (b), the accuracy is near chance (0.5), suggesting hidden states provide limited insight into prompt quality.

• **Model Preference.** We then delve into the model's ability to perceive the quality of prompts. We begin by giving the model two prompts and asking it to choose the one that would enable it to generate a more helpful, accurate, and comprehensive response. Then, we evaluate the model in two ways: (*i*) by pairing all paraphrases and asking the model's preference, and (*ii*) by only inquiring about the model's preference between the best and worst prompts.

In all experiments, we test both arrangements of the given pair of prompts to eliminate any bias that may arise from their positioning. The metric we report is the proportion of times the model selects the prompt that leads to its superior performance. As shown in Table 3, given all pairs of paraphrases, the accuracy of the Llama family models and ChatGPT hovers around 50%, with Llama-2-70B-chat slightly higher at 53.02%. When dealing with worst-best pairs, although the performance of all models, except for Llama-7B-chat, slightly improved, the highest score is only 58.92, suggesting that the models are largely unable to perceive the impact of the given prompts on their own performance.

| Model | All Pairs | Worst-Best |
|---|---|---|
| Llama-2-7B-chat | 50.05 | 49.73 |
| Llama-2-13B-chat | 49.84 | 50.57 |
| Llama-2-70B-chat | **53.02** | **58.92** |
| ChatGPT | 50.16 | 51.00 |

Table 3: In the model preference experiment, the model is tested with all paraphrase pairs (All Pairs) or best and worst prompt pairs (Worst-Best). We report the proportion of times the model prefers the prompt that leads to its better performance.

**Discussion** Our explorations over prompt perplexity, Min-k% Prob, hidden states, and model preference show that **it is very challenging to identify the worst prompt in advance even with the access to the model.** Note that this difficulty is often overlooked in previous studies as they primarily focus on task-level instructions and assume a corresponding test set for prompt selection. Specifically, they obtain the generation results on the test set and directly measure the downstream task performance. However, this is not possible for diverse real-world user queries, which further underscores the importance of the case-level setup in RobustAlpacaEval.

## 5   Improving Worst Prompt Performance

As discussed in §2, prompt engineering and prompt consistency approaches are commonly employed to improve model performance on arbitrary prompts. In this section, we investigate the effectiveness of applying these techniques to improve the worst prompt performance.

**Prompt Engineering** Most existing prompt engineering methodologies fall short in tackling the worst-performance issue, due to the lack of a test set for a single unlabeled prompt. Therefore, we explore two alternative strategies: (*i*) allowing the model to refine the prompts itself, and (*ii*) implementing a voting-based generation process that takes into account all paraphrases.

• **Self-refinement.** First, we allow the model to rewrite the given prompts according to its own preferences, making them more conducive to generating better responses.

• **Voting.** Another solution is to let the model perform voting-based generation based on all prompts, which can be mathematically represented as:

$$P(y_i|X, y_{<i}) = \mathbb{E}_{x \in X} P(y_i|x, y_{<i}) \tag{6}$$

where $X$ represents the set of paraphrases.

| Method | Orig. Perf. ↑ | Worst Perf. ↑ | Best Perf. ↑ | Avg. Perf. ↑ | Standard Dev. ↓ |
|---|---|---|---|---|---|
| *Llama-2-7b-chat* | | | | | |
| Raw | 25.61 | 5.42 | 43.54 | 19.52 | 13.32 |
| Self-refinement | 10.09(-15.52) | 1.05(-4.37) | 27.38(-16.16) | 9.48(-10.04) | 8.72(-4.60) |
| Voting | 22.35(-3.26) | 22.35(+16.93) | 22.35(-21.19) | 22.35(+2.83) | - |
| Distillation | 18.29(-7.32) | 3.89(-1.53) | 40.27(-3.27) | 19.31(-0.21) | 12.72(-0.60) |
| *Llama-2-13b-chat* | | | | | |
| Raw | 27.48 | 4.83 | 52.05 | 23.97 | 16.25 |
| Self-refinement | 12.02(-15.46) | 1.32(-3.51) | 31.40(-20.65) | 10.82(-13.15) | 10.46(-5.79) |
| Voting | 17.26(-10.22) | 17.26(+12.43) | 17.26(-34.79) | 17.26(-6.71) | - |
| Distillation | 25.90(-1.58) | 5.99(+1.16) | 47.78(-4.27) | 22.09(-1.88) | 14.30(-1.95) |
| *Llama-2-70b-chat* | | | | | |
| Raw | 32.23 | 9.38 | 54.86 | 29.18 | 15.61 |
| Self-refinement | 13.80(-18.43) | 1.02(-8.36) | 49.80(-5.06) | 15.65(-13.53) | 17.33(+1.72) |
| Voting | 31.36(-0.87) | 31.36(+21.98) | 31.36(-23.50) | 31.36(+2.18) | - |
| Distillation | 29.30(-2.93) | 7.99(-1.39) | 50.15(-4.71) | 26.44(-2.74) | 14.83(-0.78) |

Table 4: Model performance after prompt engineering (Self-refinement and Voting) and prompt consistency regularization (Distillation). The red numbers indicate a decrease in performance, while the green ones represent an improvement.

**Prompt Consistency** Drawing inspiration from the swarm distillation method proposed by Zhou et al. (2022b), we implement an unsupervised consistency regularization strategy that encourages the model's predictions for various paraphrases to converge.

• **Swarm Distillation.** To construct the training data, we utilize the SFT set (10k prompts) from the alpaca-farm dataset, and collect paraphrases based on the strategy outlined in Section 3.1. For all prompts in any given case, we sample $x_i$ and $x_j$ from them and guide the output of $x_j$ based on the model's output for $x_i$, thereby training the model in an unsupervised manner. To devise an unsupervised model selection criterion, we define the following metric:

$$C(X,Y) = \mathbb{E}_{Y_j \in Y} \sqrt{\frac{1}{N} \sum_{i=1}^{N} \left( S_{ij} - \bar{S_{ij}} \right)^2} \tag{7}$$

$$S_{ij} = \frac{1}{|Y_j|} \sum_{k=1}^{|Y_j|} \log P(Y_{j,k}|X_i, Y_{j,<k}) \tag{8}$$

where $\bar{S_{ij}}$ is the mean of $S_{ij}$ over j. The value of $C(X,Y)$ ranges from 0 to $+\infty$, reflecting the degree of consistency in the model's log confidence of generating outputs ($Y$) based on given instructions ($X$), with 0 indicating totally consistent. We obverse that the consistency always decreases at first, followed by increasing, which aligns with the findings in Zhou et al. (2022b). We train models based on LoRA and stop training at the checkpoint from which $C(X,Y)$ starts to increase.

**Results** Table 4 presents several key findings: (*i*) Model performance significantly declines when subjected to self-refined prompts, as demonstrated by Llama-2-7/13/70B-chat model's average performance, which dropped by 10.04%, 13.15%, and 13.53%, respectively. This trend suggests that direct prompt engineering without reference to test data can lead to degraded model behaviors. (*ii*) When applying the voting method, the model's best performance significantly decreases, and the average performance may also be affected (a 6.71% drop for Llama-2-13b-chat). However, it effectively improves the worst performance, for instance, boosting the Llama-2-70B-chat model by 21.98, surpassing its average performance. We believe this method is particularly effective in scenarios where the lower bound of model performance is of concern. This is because, even when faced with a less-than-optimal prompt from the user, the paraphrasing-then-voting strategy can still deliver satisfactory results. However, this improvement comes at the cost of a several-fold increase in

computational expense. (*iii*) While swarm distillation enhances the model's performance consistency, it unfortunately results in a decrease in overall performance. This is likely because the model is over-fitting to its self-generated outputs, which could introduce noise or be of lower quality than its original training data, thereby reducing the overall performance of the model.

## 6 Conclusion

In conclusion, this paper addressed a critical gap in understanding the robustness of LLMs to prompt variations. We introduced a new benchmark that shifts the focus from task-level instructions to case-level queries. Extensive experiments on ChatGPT and six open-source LLMs, revealed the substantial performance variability across different prompts, the challenge of predicting worst prompts, and the limited efficacy of existing methods for improving the worst prompts performance. Our findings underscored the importance of continued research into prompt robustness in more realistic settings. Despite the efforts, our study is not without limitations. The range of models we examined may not be extensive enough to capture a full spectrum of insights. Additionally, alternative approaches to pinpointing worst prompts could offer deeper understanding. We advocate for future studies to delve into these areas, enhancing our collective understanding.

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

| Metric | Number of Paraphrases | | | | | | | | | | |
|---|---|---|---|---|---|---|---|---|---|---|---|
| | **1** | **2** | **3** | **4** | **5** | **6** | **7** | **8** | **9** | **10** | **11** |
| Worst | 29.18 | 21.01 | 17.48 | 15.40 | 13.95 | 12.85 | 11.95 | 11.19 | 10.53 | 9.93 | 9.38 |
| Best | 29.18 | 37.34 | 41.97 | 45.17 | 47.55 | 49.42 | 50.94 | 52.18 | 53.23 | 54.11 | 54.86 |
| Best - Worst | 0.00 | 16.33 | 24.49 | 29.77 | 33.60 | 36.57 | 38.98 | 40.99 | 42.70 | 44.18 | 45.48 |

Table 5: The average score of the Llama-2-70B-chat model across all possible combinations of different numbers of paraphrases.

## A  Benchmark Construction

### A.1  Determining the Number of Paraphrases

Basically, the worst performance decreases and the best performance increases monotonically with the number of paraphrases. While it is impractical to list all possible paraphrases and calculate the exact values, we find that these scores converge quickly with a sufficiently large number of paraphrases. Taking Llama-2-70B as an example, we calculate the worst and best performances, along with their difference, for each combination of $n$ (ranging from 1 to 11) paraphrases of a case. Then, we report the average of these scores across all combinations. The results are presented in Table 5. As the results show, with the increase in the number of paraphrases, the average of the model's best/worst performance initially changes significantly and then levels off. For other models, we observe the same trend of changes. To balance the evaluation efficiency and robustness, we decided to construct 10 paraphrases for each query.

### A.2  Instruction for Paraphrasing

The few-shot instruction we used for paraphrasing is as follows:

*Your task is to generate 10 unique paraphrases for the given query, ensuring that the meaning of each paraphrase remains consistent, and the structure is significantly altered. Do not introduce any new information that isn't present in the original query, and avoid omitting any crucial information from the original query, particularly any specific requirements about the output content, style, format, options, or any numbers or data. The responses to your paraphrases and the original query should be identical. If the original query contains quoted or referenced content, such as 'rewrite the following sentence: <reference>' and those enclosed in '

', you should retain this referenced content in your paraphrases and rephrase the rest of the text. Your task is not to answer the query, but solely to rephrase it. Please provide unique and creative rephrasing. Please note that you should provide exactly 10 paraphrases, not less. Output each paraphrase separated by "||||".*
*The examples provided below are for illustrative purposes only. In the examples, only one paraphrase is provided and the other nine are omitted for brevity. However, in your output, you should provide all 10 paraphrases.*

*Examples:*
*Query:*
*"Why is it that only proteins, carbohydrates and fats are deemed to have caloric value?
Just to clarify - consider the hypothetical situation below:
100g of 'carbohydrates' = 400kCal
100g of 'protein' = 400kCal
100g of 'fat' = 900kCal
100g of 'vitamins, minerals and/or other electrolytes' = ?"*
***Output:*** *Why are proteins, carbohydrates, and fats the only substances considered to have caloric value? To provide further clarification, let's consider the hypothetical scenario below: If 100g of carbohydrates provide 400kCal, 100g of protein provides 400kCal, and 100g of fat provides 900kCal, what would be the caloric value of 100g of vitamins, minerals, and/or other electrolytes? |||| <other nine paraphrases are omitted for brevity>*

***Query:*** *"I have this text: 

We use professional processes. Use the photo delivery system, check the quality and send it to the customer immediately. We started editing digital photos from the first wave of 2000. Started with editing personal photos, and then spread our passion by editing wedding photos or restoring old photos for loved ones, friends and family. friends and cus-*

*tomers. 2005 started a local photo shop and started commercial photo editing, product photography for manufacturers. 2007 implemented online projects on freelancer platforms.

Rewrite it in professional style to tell a touching story."*
**Output:** *I possess this text: 

We use professional processes. Use the photo delivery system, check the quality and send it to the customer immediately. We started editing digital photos from the first wave of 2000. Started with editing personal photos, and then spread our passion by editing wedding photos or restoring old photos for loved ones, friends and family. friends and customers. 2005 started a local photo shop and started commercial photo editing, product photography for manufacturers. 2007 implemented online projects on freelancer platforms.

Rephrase it in a professional manner to narrate a moving story.*

**Query:** *"I want you to act as a Tiktok Ads Keywords generator. Your task is to come up with keywords that can be used in Tiktok advertising campaigns. The keywords should be relevant to the target audience and the product or service being advertised. Your responses should not include more than 5 keywords per request."*
**Output:** *Your role will be to serve as a TikTok Ads Keywords generator. Your objective is to generate relevant keywords that can be utilized in TikTok advertising campaigns. These keywords should effectively target the desired audience and align with the product or service being promoted. Please provide no more than five keywords per request in your responses.*

*Now, please generate 10 paraphrases for the given query.*

**Query:** $\{input\_query\}$
**Output:**

# B Identifying the Worst Prompts

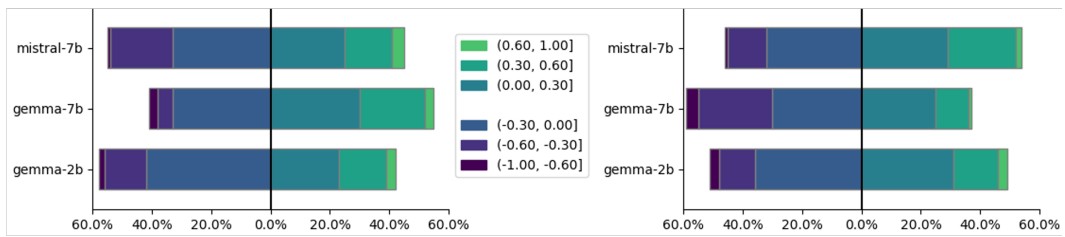

Figure 6: Distribution of Pearson correlation coefficients between model performance and prompt perplexity (left) and prompt's Min-K% Prob (right) for Gemma family models and Mistral-7B model across all cases.

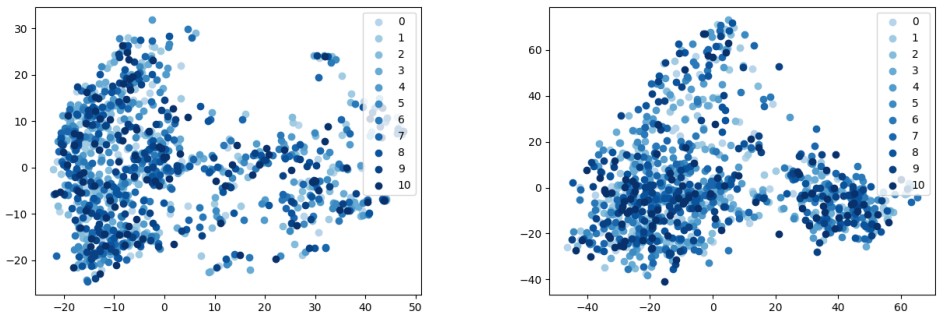

Figure 7: Visualization of the hidden states from Llama-2-13B-chat (left) and Llama-2-70B-chat (right) models using 2-dimensional PCA. The color gradient, from light to dark, represents the ranking of model performance on each case's 11 prompts, from low to high.

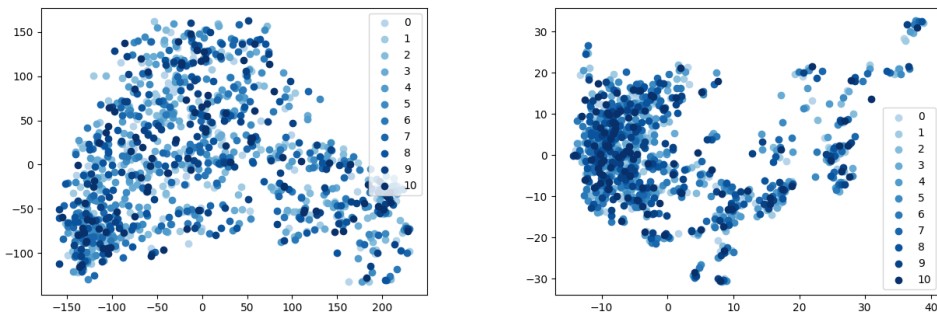

Figure 8: Visualization of the hidden states from Mistral-7B (left) and Gemma-2B (right) models using 2-dimensional PCA. The color gradient, from light to dark, represents the ranking of model performance on each case's 11 prompts, from low to high.

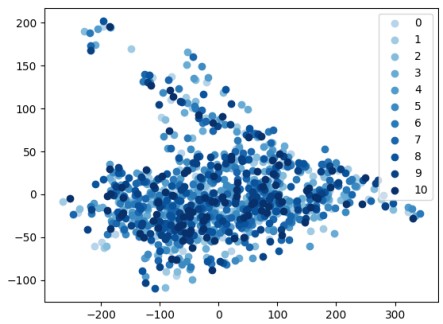

Figure 9: Visualization of the hidden states from Gemma-7B model using 2-dimensional PCA. The color gradient, from light to dark, represents the ranking of model performance on each case's 11 prompts, from low to high.

**Original Instruction (Score: 0.9922)**
On the basis of the subject of the email, determine whether the email should be treated as spam or not.
Dermatologists don't like her!
**Output:**
Based on the subject of the email, it is likely that the email should be treated as spam. The use of the phrase "Dermatologists don't like her" is likely to be a sensationalized or misleading claim ...

**Paraphrase (Score: 0.0005)**
Examine the email topic to ascertain if it should be categorized as spam. Subject: Dermatologists don't like her!
**Output:**
This email topic does not appear to be spam. Here's why: 1. Relevance: ... It is not a generic or random topic that is often associated with spam emails. 2. Urgency: ... 3. Personalization: The email is addressed to a specific person, which suggests that it may be a legitimate message rather than a mass email blast. 4. Grammar and spelling: The email appears to be well-written and free of obvious grammatical and spelling errors, which is another indicator that it may be a legitimate message.
Based on these factors, it is unlikely that this email should be categorized as spam.

Figure 10: Example for sensitive cases. Despite providing instructions (other paraphrases are emitted for brevity) with the same semantics, the output from Llama2-7b-chat is completely contradictory. Furthermore, there are numerous inaccurate expressions (highlighted in red) in the underperforming response.

