# OpenReview forum: "On the Worst Prompt Performance of Large Language Models"
_NeurIPS.cc/2024/Conference — NeurIPS 2024 poster_

### Official Review · Reviewer_Wkdh · 2024-07-02

**Soundness:** 3
**Presentation:** 3
**Contribution:** 3
**Rating:** 6
**Confidence:** 3

**Summary:**

The authors propose a new benchmark to study the robustness of LLMs to prompt variations. Different from previous work, this paper mainly focuses on semantically equivalent prompts rather than taks-level instructions. The experiments demonstrate that many popular LLMs are sentitive to the form of prompts. Moreover, it is difficult to detect the worst prompt and improve the worst prompt performance.

**Strengths:**

1. Research Value: The paper tackles a highly relevant problem that has substantial implications for the practical deployment of LLMs. The issue of prompt sensitivity is critical and the paper provides valuable insights into this domain.

2. Clarity of Motivation: The authors have effectively articulated the motivation behind their work. The introduction and the problem statement are clear, setting the stage for the reader to understand the importance of the study.

**Weaknesses:**

1. Lack of Solution: While the paper adeptly identifies the problem of prompt variability and its impact on LLM performance, it falls short of proposing an effective solution to mitigate this issue. The research would benefit from the inclusion of potential remedies or strategies to enhance prompt robustness.

2. Inconsistency in Description: There appears to be a discrepancy between the textual description and the visual representation (Figure 1) regarding the focus on task-level instructions versus case-level inputs. Clarification on this point is needed to avoid any confusion for the readers.

3. Limited Scope of Experimentation: The experiments, while extensive, do not include testing on more advanced models such as GPT4. The performance of the proposed benchmark on state-of-the-art models would provide a more comprehensive understanding of the robustness of the latest LLMs.

4. Language Specificity: The study seems to be limited to English language prompts. Given the global application of LLMs, it would be beneficial to extend the experiments to include other languages to assess the generalizability of the findings.

5. Prompt Diversity Validation: The paper does not explore whether incorporating a diverse set of prompts in the SFT stage could potentially improve model's worst prompt performance. An analysis of this nature would add depth to the paper's contribution.

**Questions:**

See Weaknesses.

**Limitations:**

The authors have adequately addressed the limitations.

---

> ### Author Rebuttal · Authors · 2024-08-07
>
> Thank you for the positive feedback. We appreciate that you acknowledge the thorough evaluations and insightful guidelines in our paper. We provide point-to-point responses to address your concerns as follows:
>
> **[W1]: potential remedies or strategies to enhance prompt robustness.**
>
> **[A1]**: Thank you for your insightful questions. We explored various methods to identify the worst prompt and improve the model's robustness. Unfortunately, the investigated methods are not effective enough to mitigate the problem. We would like to clarify that the spotting of this problem is also a valuable contribution as it has been neglected by previous work. As a consequence, previous solutions fail to address the problem.
>
> The contribution of our work lies, first and foremost, in pioneering the shift from task-level instructions to case-level queries and capitalizing on the concept of worst prompt performance. Our benchmark provides a general test for researchers to evaluate a model's ability to provide stable responses for real-world users before deploying LLMs. Our comprehensive experiments highlight the great challenge of identifying the worst prompt and improving their performance in this realistic setting. Our results thoroughly examine existing efforts in reducing model sensitivity to prompt variations and clearly delineate their limitations.
>
> We anticipate other attempts like prompt engineering techniques, model retraining with paraphrases, and integrating external knowledge sources can potentially address this problem, and we leave them for further work.
>
> **[W2]: There appears to be a discrepancy between the textual description and the visual representation (Figure 1) regarding the focus on task-level instructions versus case-level inputs. Clarification on this point is needed to avoid any confusion for the readers.**
>
> **[A2]**: Could you please specify where you perceive a discrepancy in the statement? Currently, we use task-level instructions and case-level inputs to represent the two parts into which previous studies decompose a complete case-level query. In our setting, we study the variations of case-level queries and do not distinguish between instructions and inputs.
>
> **[W3]: The experiments, while extensive, do not include testing on more advanced models such as GPT4. The performance of the proposed benchmark on state-of-the-art models would provide a more comprehensive understanding of the robustness of the latest LLMs.**
>
> **[A3]**: We selected models based on their popularity and advanced nature. These models represent the state-of-the-art open-source LLMs and are widely adopted in existing research. Due to the limited budget for GPT4, we did not test the performance of the GPT4 model.
>
> **[W4]: The study seems to be limited to English language prompts. Given the global application of LLMs, it would be beneficial to extend the experiments to include other languages to assess the generalizability of the findings.**
>
> **[A4]**: Thank you for your suggestion. We anticipate that other languages would face similar, possibly more severe, issues given the current LLMs are mostly trained in English. We will investigate whether other languages face the same issues in future work.
>
> **[W5]: The paper does not explore whether incorporating a diverse set of prompts in the SFT stage could potentially improve a model's worst prompt performance. An analysis of this nature would add depth to the paper's contribution.**
>
> **[A5]**: In fact, current state-of-the-art LLMs (those included in our study) have been extensively fine-tuned using a huge number of diverse prompts. Nevertheless, our experiment results show that they are still not robust enough to address our benchmark. Additionally, the swarm distillation technique we used in our experiments [Line 298-310] is related to the SFT you mentioned. It works by aligning the outputs generated by the model for different paraphrases to improve the consistency of the model's results. Our results indicate that the benefits of this method are not significant.

---

> ### Author Response · Authors · 2024-08-12
>
> Thank you again for your positive feedback! We are grateful for your recognition of the value of the problem we study and the insights of our findings!
>
> We hope our clarifications addressed your comments and we would like to inquire if you have any further questions or require additional information. Would you kindly be open to increasing your score? We are eager to provide any necessary support and continue the dialogue.

---

> > ### Comment · Reviewer_Wkdh · 2024-08-13
> >
> > The authors' response has addressed most of my concerns. I will raise my score to 6.

---

> > > ### Author Response · Authors · 2024-08-14
> > >
> > > Thank you for your valuable feedback and the consideration of raising the score after our discussion. We are pleased that our rebuttal has addressed your concerns. We will also address these concerns and incorporate all your suggestions in our paper.
> > >
> > > Thanks again for your insightful comments that help improve our work greatly!

---

### Official Review · Reviewer_4kDi · 2024-07-08

**Soundness:** 3
**Presentation:** 3
**Contribution:** 2
**Rating:** 6
**Confidence:** 3

**Summary:**

This paper introduces a new benchmark, ROBUSTALPACAEVAL, that contains semantically equivalent queries of diverse real-world tasks, and uses it to conduct extensive experiments on ChatGPT and six open-source LLMs. It highlights variability in model performance and difficulties in predicting the worst prompt.

**Strengths:**

- This work introduces a new benchmark focused on diverse real-world user queries rather than task-level instructions, which more closely mirrors real-world scenarios.
- It conducts comprehensive evaluations across several LLMs and showcases variability in model performance across different prompts.
- They tried several methods to improve the worst prompt and show the limitations of current method.

**Weaknesses:**

- The methodology utilizes gpt4_turbo as the evaluator and the reference model for outputs, which might introduce biases. A discussion of this potential bias should be included.
- The ROBUSTALPACAEVAL benchmark contains 10 paraphrases per given query. It’s unclear how performance disparities between the worst and best prompts would change with an increased number of paraphrases. This might also affect the robustness of cases with significant performance gaps.
- While the study explores several LLM families including Llama, Gemma, and Mistral, it includes only one version of ChatGPT and does not specify which version. Whether the prompts are uniform across different ChatGPT versions is unknown.

**Questions:**

- How is prompt perplexity calculated?
- Are there any common features among the worst-performing prompts within the same model family? The paper points out the difficulty in predicting the worst prompts, both model-independently and with model access. Given that the overlap rate of model-agnostic worst-4 prompts reaches about 40% in the Gemma family, an analysis comparing the features of the worst and best prompts could be insightful.

**Limitations:**

The paper includes a limited selection of LLMs, and the challenge of effectively improving the worst-performing prompts remains unresolved.

---

> ### Author Rebuttal · Authors · 2024-08-07
>
> Thank you for your thorough review and for highlighting the strengths of our work. We appreciate your constructive feedback requiring further clarifications. Below, we address each of your points in detail.
>
> **[W1]: The methodology utilizes gpt4_turbo as the evaluator and the reference model for outputs, which might introduce biases. A discussion of this potential bias should be included.**
>
> **[A1]**: Thank you for your insightful questions. The evaluation metric we use is introduced by the AlpacaEval 2.0 benchmark, which has been proven to achieve high consistency to human evaluations and has been widely adopted by the LLM research community. Additionally, following the common practice, all models are first compared with gpt4_turbo then we compare their win rates. Therefore, if the assumed bias is present, all models are affected, ensuring a fair comparison. We will include the above discussion in our revisions.
>
> **[W2]: The ROBUSTALPACAEVAL benchmark contains 10 paraphrases per given query. It’s unclear how performance disparities between the worst and best prompts would change with an increased number of paraphrases. This might also affect the robustness of cases with significant performance gaps.**
>
> **[A2]**:  Indeed, the worst/best performance decreases/increases monotonically with the number of paraphrases (n). While it is impractical to list all possible paraphrases and calculate exact values, we find that these scores converge quickly with a sufficiently large number of paraphrases. Taking Llama-2-70B as an example, we calculated the worst/best performance and their difference from n=1 to 11 as below (We report the averaged score across all possible combinations and the standard deviations). For other models, we observe the same trend of changes. To balance the evaluation efficiency and robustness, we decided to construct 10 paraphrases for each query.
>
> | Metric | n=1 | n=2 | n=3 | n=4 | n=5 | n=6 | n=7 | n=8 | n=9 | n=10 | n=11 |
> | :----: | :----: | :----: | :----: | :----: | :----: | :----: | :----: | :----: | :----: | :----: | :----: |
> | Worst | 29.18(1.42) | 21.01(1.25) | 17.48(1.26) | 15.4(1.14) | 13.95(1.02) | 12.85(0.92) | 11.95(0.83) | 11.19(0.73) | 10.53(0.59) | 9.93(0.42) | 9.38(0.0) |
> | Best | 29.18(1.42) | 37.34(1.82) | 41.97(1.77) | 45.17(1.64) | 47.55(1.47) | 49.42(1.28) | 50.94(1.08) | 52.18(0.88) | 53.23(0.67) | 54.11(0.43) | 54.86(0.0) |
> | Best - Worst | 0.0(0.0) | 16.33(2.48) | 24.49(2.54) | 29.77(2.32) | 33.6(2.06) | 36.57(1.79) | 38.98(1.53) | 40.99(1.25) | 42.7(0.96) | 44.18(0.63) | 45.48(0.0) |
>
> **[W3]: While the study explores several LLM families including Llama, Gemma, and Mistral, it includes only one version of ChatGPT and does not specify which version. Whether the prompts are uniform across different ChatGPT versions is unknown.**
>
> **[A3]**: The version of ChatGPT we utilize is gpt-3.5-turbo-1106.
>
> **[Q1]: How is prompt perplexity calculated?**
>
> **[A4]**: We calculate the average log probability of all the tokens in the prompt and take the inverse of the exponential of the average log probability as the perplexity of the prompt.
>
> **[Q2]: Are there any common features among the worst-performing prompts within the same model family? The paper points out the difficulty in predicting the worst prompts, both model-independently and with model access. Given that the overlap rate of model-agnostic worst-4 prompts reaches about 40% in the Gemma family, an analysis comparing the features of the worst and best prompts could be insightful.**
>
> **[A5]**: We conducted extensive manual analysis and found no discernible difference between the worst- and best-performing prompts that humans can perceive. We observe a high overlap rate of worst prompts among models within the same family, which we hypothesize stems from their shared preferences shaped by factors like the distribution of training data (which is not fully accessible to us), indicating a need for a deeper understanding and knowledge of the LLMs.

---

> ### Author Response · Authors · 2024-08-12
>
> Thank you again for your positive feedback! We are grateful for your recognition of the value of the problem we study and the insights of our findings!
>
> We hope our clarifications addressed your comments and we would like to inquire if you have any further questions or require additional information. Would you kindly be open to increasing your score? We are eager to provide any necessary support and continue the dialogue.

---

> > ### Comment · Reviewer_4kDi · 2024-08-13
> >
> > Thanks to the authors for the detailed response. The rebuttal has addressed most of my concerns. I will keep a positive score. Good luck!

---

> > > ### Author Response · Authors · 2024-08-14
> > >
> > > Thank you for your valuable feedback and for firmly backing the acceptance of our paper. We are glad our rebuttal has addressed your concerns. We promise to incorporate all your suggestions and tackle any remaining issues in our updated paper.
> > >
> > > Thank you again for your insightful advice and continued support.

---

### Official Review · Reviewer_H71g · 2024-07-11

**Soundness:** 3
**Presentation:** 4
**Contribution:** 3
**Rating:** 6
**Confidence:** 3

**Summary:**

In this paper, the author(s) propose a benchmark for prompt performance of large language models. In particular, the author(s) leverage GPT4 to generate variants of prompts and hence constitute a dataset. Afterwards, the author(s) evaluate the performance of these generated prompts, and explore the identification and improvement of worst cases.

**Strengths:**

- This study acknowledges the variation and diversity of prompts in real-world, and adds value to the prompt engineering for LLMs.
- The proposed benchmark is applied to multiple LLMs. Thorough comparison analysis is conducted.
- The paper is well-organized, and the language is technical yet understandable for readers with domain knowledge.

**Weaknesses:**

- The methodology and result analysis of this study can be elaborated.
- Result visualization can be improved for better readability.

**Questions:**

- Section 2: A related work can be added and analysed. Schulhoff, S., Ilie, M., Balepur, N., Kahadze, K., Liu, A., Si, C., ... & Resnik, P. (2024). The Prompt Report: A Systematic Survey of Prompting Techniques. arXiv preprint arXiv:2406.06608.
- Please elaborate the motivation and result analysis for exploring the hidden states in Section 4.2.
- I wonder if the author(s) consider sharing the dataset used in this project?
- The author(s) can consider specifying the selection criteria of LLMs used in experiments.
- Please consider using bold in tables to highlight the noteworthy data.
- It will be interesting to summarize the features/patterns of worst prompts for different LLMs.
- Please adjust the place of tables and figures to improve readability. For instance, Figure 3 is on Page 6 while it is described on Page 7.

**Limitations:**

- The author(s) note that the research can be extensive by including more LLMs, and applying alternative approaches to identify worst prompts.

---

> ### Author Rebuttal · Authors · 2024-08-07
>
> Thank you for your valuable insights of our work. We appreciate your thoughtful feedback, and we would like to address your concerns as follows:
>
> **[W1]: The methodology and result analysis of this study can be elaborated.**
>
> **[A1]**: We appreciate the reviewer's feedback on the need for a more detailed explanation of our methodology and result analysis. We will expand on the rationale behind our method and provide a more thorough interpretation of the results when more space is allowed.
>
> **[W2]: Result visualization can be improved for better readability.**
>
> **[A2]**: We will work on enhancing the clarity and effectiveness of our visual presentations, ensuring that key findings are prominently displayed.
>
> **[Q1]: Section 2: A related work can be added and analysed. Schulhoff, S., Ilie, M., Balepur, N., Kahadze, K., Liu, A., Si, C., ... & Resnik, P. (2024). The Prompt Report: A Systematic Survey of Prompting Techniques. arXiv preprint arXiv:2406.06608.**
>
> **[A3]**: Thank you for your suggestion. This survey has a nice summary of existing prompting techniques and we are happy to include this work in our next version.
>
> **[Q2]: Please elaborate the motivation and result analysis for exploring the hidden states in Section 4.2.**
>
> **[A4]**: As demonstrated by many previous works, hidden states can reflect a model's perception and preferences. Our motivation for studying hidden states is to determine whether there is a correlation between the model's own encoding of the prompt and its performance.
>
> We conducted two experiments [Line 243-259]. The PCA results showed no clear boundary between the representations of prompts with different performance levels; in the other experiment, we trained a classifier to predict which prompt the model performs better on based on the hidden states, and found that the trained classifier only achieved about 50% accuracy on the test set. The results of both experiments indicate that it is difficult to predict the model's performance based on the model's representation of the prompt.
>
> **[Q3]: I wonder if the author(s) consider sharing the dataset used in this project?**
>
> **[A5]**: We have released the dataset used in this project. We will ensure that the dataset is properly documented.
>
> **[Q4]: The author(s) can consider specifying the selection criteria of LLMs used in experiments.**
>
> **[A6]**: We selected models based on their popularity and advanced nature. These models represent the state-of-the-art open-source LLMs and are widely adopted in existing research.
>
> **[Q5]: Please consider using bold in tables to highlight the noteworthy data.**
>
> **[A7]**: We will implement this to make the most significant findings and differences between conditions more apparent at a glance.
>
> **[Q6]: It will be interesting to summarize the features/patterns of worst prompts for different LLMs.**
>
> **[A8]**: We conducted extensive manual analysis and found no discernible difference between the worst- and best-performing prompts that humans can perceive. We observe a high overlap rate of worst prompts among models within the same family, which we hypothesize stems from their shared preferences shaped by factors like the distribution of training data (which is not fully accessible to us), indicating a need for a deeper understanding and knowledge of the LLMs.
>
> **[Q7]: Please adjust the place of tables and figures to improve readability. For instance, Figure 3 is on Page 6 while it is described on Page 7.**
>
> **[A9]**: We will review and adjust the positioning of these elements to ensure they appear as close as possible to their first reference in the text.

---

> ### Author Response · Authors · 2024-08-12
>
> Thank you again for your positive feedback! We are grateful for your recognition of the value of the problem we study and the insights of our findings!
>
> We hope our clarifications addressed your comments and we would like to inquire if you have any further questions or require additional information. Would you kindly be open to increasing your score? We are eager to provide any necessary support and continue the dialogue.

---

### Official Review · Reviewer_7q8k · 2024-07-13

**Soundness:** 3
**Presentation:** 3
**Contribution:** 2
**Rating:** 5
**Confidence:** 4

**Summary:**

This paper proposes a new benchmark, RobustAlpacaEval, a benchmark with semantically equivalent queries. The authors evaluate performance as the worst performance across all the prompts and show that many prompt consistency methods have a limited improvement on this benchmark. The paper also shows that it is difficult to predict which prompts will have a poor performance.

**Strengths:**

- The paper has a variety of key findings (Section 3.2). First, the authors find that there is a large gap between worst and best performance. Second, the authors find that scaling does not help robustness. Third, the authors find that there is little agreement between various models on which prompts are harder and/or easier.
- The experiments are thorough and well conducted.
- The findings open up doors for future research and efforts to make models more robust.

**Weaknesses:**

- The variance of a model to different prompts has been studied several times in the literature (for example, see [1]), and it is unclear to me how a benchmark measuring worst-case performance would be used. I would appreciate if the authors provided more clarity on how they envision that this work could shed insight and facilitate future efforts.
- In Section 5, only Llama models are considered, which makes it unclear whether the insights discovered also apply to other models or are unique to Llama.
- The paper would be much stronger if the authors proposed a method to mitigate the phenomenon demonstrated in the paper, or if the authors gave a detailed explanation of why the identified phenomenon occurs.

[1] https://arxiv.org/abs/2401.03729

**Questions:**

- Can the authors suggest some potential directions for resolving the issue raised in the paper?
- When creating a benchmark like this, the number of prompts can be increased arbitrarily, and the accuracy could be extremely low if the right prompts are chosen. Similarly, one could create an adversarial setting to test the robustness of each model. Why did you design the benchmark in this way?
- How do the authors envision that this benchmark be used? For example, should all future models be forced to evaluate on RobustAlpacaEval? Because there are many more prompts for each sample, this would lead to a multiplicative factor on the cost for inference.

**Limitations:**

Yes

---

> ### Author Rebuttal · Authors · 2024-08-07
>
> We appreciate your in-depth review and the recognition of our novelty, the importance, and the empirical results. We understand your concerns and address them with the following clarifications.
>
> **[W1]: relation to prior work &  future impact of our work.**
>
> **[A1]**: Thank you for your insightful questions. Our work significantly differs from most previous research in two aspects. First, we focus on the model's robustness when faced with prompts that have the same meaning, unlike methods that introduce intentional perturbations, potentially affecting the semantics (e.g.,  [1] focuses on a few special prompt perturbations, including changes in output format, jailbreaks, and tipping strategies, which can alter the semantics of the original prompts). Second, We investigate the impact of paraphrasing case-level user queries, which is more aligned with real-world needs compared to past studies that focused solely on task-level instructions. We will add the discussion about [1] in our revisions.
>
> The significance of our work lies, first and foremost, in pioneering the shift from task-level instructions to case-level queries and capitalizing on the concept of worst prompt performance. Our benchmark provides a general test for evaluating a model's ability to provide stable responses for real-world users before deploying LLMs. Our experiments highlight the great challenge of identifying the worst prompt and improving their performance in this realistic setting. Our results thoroughly examine existing efforts in reducing model sensitivity to prompt variations and clearly delineate their limitations.
>
> **[W2]: results for other models.**
>
> **[A2]**: Thanks for your constructive suggestion. We would like to emphasize that we have already presented the results using LLMs of different scales in Table 4. For other model families, we observed similar phenomena and omitted the discussion for saving space. For example, the results on Gemma-1.1-2b-it are given in the following table. We can see that the general conclusions (Line 311 to 325) regarding the effect of different methods (self-refinement, voting, and distillation) are consistent with those of the llama family.
>
> | Method | Orig. Perf.&uarr; | Worst Perf.&uarr; | Best Perf.&uarr; | Avg. Perf.&uarr; | Standard Dev.&darr; |
> | :----: | :----: | :----: | :----: | :----: | :----: |
> | Raw | 16.32 | 4.42 | 36.6 | 15.27 | 11.78 |
> | Self-Refinement | 6.75(-9.57) | 0.03(-4.39) | 18.44(-18.16) | 4.55(-10.72) | 6.06(-5.72) |
> | Voting |14.30(-2.02) | 14.30(+9.88) | 14.30(-22.30) | 14.30(-0.97) | - |
> | Distillation | 13.08(-3.24) | 1.67(-2.75) | 31.36(-5.24) | 11.28(-3.99) | 9.93(-1.85) |
>
> **[W3]: Mitigation or explanation of the phenomenon.**
>
> **[A3]**: We explored various methods to identify the worst prompt and improve the model's robustness. Unfortunately, all these methods are not effective enough to mitigate the problem. In addition to the automatic analysis presented in the paper, we conducted extensive manual analysis on the worst prompt and found no discernible difference between the worst- and best-performing prompts that humans can perceive. We believe that understanding and addressing this problem requires a deeper understanding and knowledge of the LLMs. For instance, we observe a high overlap rate of worst prompts among models within the same family, which we hypothesize stems from their shared preferences shaped by factors like the distribution of training data (which is not fully accessible to us).
>
> **[Q1]: Potential directions for resolving the issue.**
>
> **[A4]**: Our research reveals that existing efforts for reducing model sensitivity are not as effective in our setup. We are actively exploring strategies such as prompt engineering techniques, model retraining with paraphrases, and integrating external knowledge sources to alleviate this issue.
>
> **[Q2]: the number of prompts & designing the benchmark.**
>
> **[A5]**: The worst/best performance decreases/increases monotonically with the number of paraphrases (n). While it is impractical to list all possible paraphrases and calculate exact values, we find that these scores converge quickly with a sufficiently large number of paraphrases. Taking Llama-2-70B as an example, we calculated the worst and best performance from n=1 to 11 as below (We report the averaged score across all possible combinations and the standard deviations). For other models, we observe the same trend of changes. To balance the evaluation efficiency and robustness, we decided to construct 10 paraphrases for each query.
>
> | Metric | n=1 | n=2 | n=3 | n=4 | n=5 | n=6 | n=7 | n=8 | n=9 | n=10 | n=11 |
> | :----: | :----: | :----: | :----: | :----: | :----: | :----: | :----: | :----: | :----: | :----: | :----: |
> | Worst | 29.18(1.42) | 21.01(1.25) | 17.48(1.26) | 15.4(1.14) | 13.95(1.02) | 12.85(0.92) | 11.95(0.83) | 11.19(0.73) | 10.53(0.59) | 9.93(0.42) | 9.38(0.0) |
> | Best | 29.18(1.42) | 37.34(1.82) | 41.97(1.77) | 45.17(1.64) | 47.55(1.47) | 49.42(1.28) | 50.94(1.08) | 52.18(0.88) | 53.23(0.67) | 54.11(0.43) | 54.86(0.0) |
>
> It is noteworthy that the construction of our benchmark is not tailored to any specific model but provides a general testing method to evaluate the robustness of models when faced with semantically equivalent but differently phrased instructions. As the experimental results show, different models exhibit similar fluctuations in their performance, while the worst prompts for each model are unique.
>
> **[Q3]: The usefulness of the benchmark.**
>
> **[A6]**: We envision ROBUSTALPACAEVAL as a tool in the broader toolkit for evaluating LLMs. Models can be evaluated on this benchmark to ensure they meet a minimum standard of robustness. While the increased number of prompts raises computational costs, this is balanced by the critical insight it provides into model reliability.

---

> ### Author Response · Authors · 2024-08-12
>
> Thank you again for your positive feedback! We are grateful for your recognition of the richness of our experiments and the contribution of our research to the community!
>
> We hope our clarifications addressed your comments and we would like to inquire if you have any further questions or require additional information. Would you kindly be open to increasing your score? We are eager to provide any necessary support and continue the dialogue.

---

> > ### Comment · Reviewer_7q8k · 2024-08-13
> >
> > Thanks to the authors for their response. The provided method certainly does measure some sort of robustness metric. In my humble opinion, this level of robustness is one step above just using one prompt, but is still far away from measuring robustness against real and adversarial attacks.
> >
> > In any case, I am still supporting the acceptance of the paper, and I will keep my score, which is above the acceptance threshold.

---

> ### Author Response · Authors · 2024-08-13
>
> Thank you for your prompt reply. We fully agree with your view that "it is still far away from measuring robustness against real and adversarial attacks," and we would like to share some of our thoughts with you.
>
> Firstly, while robustness against attacks is a critical security issue, our focus is more on model stability, particularly its response to prompts with varying but equally clear expressions of the same meaning. We observe that most users do not deliberately attack models, but their distinct language styles can unconsciously influence the quality of the model's outputs.
>
> We believe that model stability is an area that deserves more attention, and the "worst prompt performance of models" we have proposed is an often-overlooked issue. Our work presents a dedicated and comprehensive analysis of measuring, predicting, and improving worst prompt performance. We hope our paper will inspire further research in this area.
>
> Lastly, we appreciate your explicit statement of **“supporting the acceptance of the paper”**. However, a borderline accept score typically indicates uncertainty and a significant chance of rejection, in contrast to a weak accept or higher score, which signals a clear intent to publish. We kindly request that you consider adjusting the score upwards.
>
> Thanks again for your insightful comments and encouraging reply.

---

### Official Review · Reviewer_y4yW · 2024-07-13

**Soundness:** 3
**Presentation:** 3
**Contribution:** 3
**Rating:** 6
**Confidence:** 4

**Summary:**

This paper studies the worst performance an LLM can have on input queries by testing on paraphrases of each query and report the original, worst, best, and average performance. It has been found that there is a large gap between the best and worst performance. It is then found that there is no particular "worst prompt" among all models or even models from the same family. Furthermore, model-dependant features like are also not useful in identifying the worst prompt. Finally, authors explored a few approaches to try to improve the "worst performance". In particular, voting seems effective in improving the lower bound at the cost of harming the upper bound.

**Strengths:**

- This paper discusses an important question to study, that is the lower bound performance of LLMs on different prompts. It is interesting to know that identifying the prompt that leads to bad performance is difficulty by looking at the prompts alone or using an LLM itself.

**Weaknesses:**

- I think the definition of the "worst" or "best" performance is not clearly stated (line 121). If I understood correctly, the worst performance will monotonically drop if we have more paraphrases of each query. This is linked to the question below and I hope that the authors can clarify on this.
- Using paraphrases to study prompt variation could introduce errors and paraphraser/LLM preference mismatch. It might be questioned that Table 1 and Table 4's average performance is almost always lower than the original performance. Does this mean that the paraphrasing process has introduced errors, domain shifts, etc?

**Questions:**

- I wonder if you could provide a clear and reproducible definition of the worst and best performance in mentioned line 121. From the writing, I think regarding whether a query is correctly responded by the model: worst=all(prompt1, prompt2, ...) and best=any(prompt1, prompt2, ...). Is this correct?
    - If this is the case, I think the more paraphrases we have, the higher we get for the best performance and vice versa for the worst performance. This might make the number of paraphrases a bit arbitrary. The worst-best gap might also be inflated.


- Throughout the paper, starting with line 15, I think the phrase "worst prompt" is not precise---we cannot call them good or bad per se unless proven. I think that you mean the prompt paraphrase that leads to the worst performance.

**Limitations:**

The paper touches on limitations like not experimenting enough models and enough techniques in finding the "worst prompt".

---

> ### Author Rebuttal · Authors · 2024-08-07
>
> Thank you for your thoughtful review and highlighting our work's strengths. We appreciate your constructive feedback on areas needing further elaboration. We address each of your points below.
>
> **[W1 & Q1]: definition of the "worst" or "best" performance & the number of paraphrases**
>
> **[A1]**: Thank you for your insightful questions. Your understanding is essentially correct. In our study, "worst" and "best" performance refer to the lowest and highest performance among all paraphrases of a query. The only correction needed is that the metric we use is based on the evaluator scoring the model's output on a continuous scale from 0 to 100. Therefore, we have: worst=min(prompt1, prompt2, …), and best=max(prompt1, prompt2, …).
>
> Indeed, the worst/best performance decreases/increases monotonically with the number of paraphrases (n). While it is impractical to list all possible paraphrases and calculate exact values, we find that these scores converge quickly with a sufficiently large number of paraphrases. Taking Llama-2-70B as an example, we calculated the worst/best performance and their difference from n=1 to 11 as below (We report the averaged score across all possible combinations and the standard deviations). For other models, we observe the same trend of changes. To balance the evaluation efficiency and robustness, we decided to construct 10 paraphrases for each query.
>
> | Metric | n=1 | n=2 | n=3 | n=4 | n=5 | n=6 | n=7 | n=8 | n=9 | n=10 | n=11 |
> | :----: | :----: | :----: | :----: | :----: | :----: | :----: | :----: | :----: | :----: | :----: | :----: |
> | Worst | 29.18(1.42) | 21.01(1.25) | 17.48(1.26) | 15.4(1.14) | 13.95(1.02) | 12.85(0.92) | 11.95(0.83) | 11.19(0.73) | 10.53(0.59) | 9.93(0.42) | 9.38(0.0) |
> | Best | 29.18(1.42) | 37.34(1.82) | 41.97(1.77) | 45.17(1.64) | 47.55(1.47) | 49.42(1.28) | 50.94(1.08) | 52.18(0.88) | 53.23(0.67) | 54.11(0.43) | 54.86(0.0) |
> | Best - Worst | 0.0(0.0) | 16.33(2.48) | 24.49(2.54) | 29.77(2.32) | 33.6(2.06) | 36.57(1.79) | 38.98(1.53) | 40.99(1.25) | 42.7(0.96) | 44.18(0.63) | 45.48(0.0) |
>
>
> **[W2]: Does the paraphrasing process introduce errors, domain shifts, etc?**
>
> **[A2]**: We would like to clarify that we manually reviewed and revised all paraphrases to ensure semantic equivalence and human-like fluency, for which errors or domain shifts are largely prevented. Regarding the relationship between the average and original performance, we consider the instances where the average performance is lower than the original to be coincidental, as there are also instances where the opposite is true (2/7 in Table 1, 5/12 in Table 4). Note that the original performance is measured only by a single paraphrase. The randomness further underscores the value of our benchmark setup.
>
> **[Q2]: Clarification on the phrase "worst prompt".**
>
> **[A3]**: Thank you for pointing this out. We intend to use "worst prompt" to refer to the paraphrase of a query that causes the model to perform the worst. We will clarify this in our next version.

---

> ### Comment · Reviewer_y4yW · 2024-08-11
>
> Thank you for the response. I think my questions are largely resolved. I have adjusted my score. Good luck.
>
> I agree with Reviewer H71g that a few other papers that have studied prompt variations and robustness can be referenced.

---

> > ### Author Response · Authors · 2024-08-12
> >
> > Thank you for your positive feedback! We are glad that our rebuttal addresses your concerns.  We will also address these concerns and incorporate all your suggestions in our paper. Thanks again for your insightful comments that help improve our work greatly.

---

### Decision · Program_Chairs · 2024-09-25

**Decision:**

Accept (poster)

**Comment:**

The paper studies the worst performance of LMMs through a new benchmark they propose as part of this study that is composed of semantically equivalent queries generated and proposes to use the lower bound of model performance that helps characterize a lower bound for model performance. The reviewers unanimously recommend its acceptance. They mention that the paper is well written, noted the importance of the problem and commended on the thoroughness of experiments. I therefore recommend its acceptance.